

# A Bayesian Approach to Infer Nitrogen Loading Rates from Crop and Landuse Types Surrounding Private Wells in the Central Valley, California

Katherine M. Ransom[1], Andrew M. Bell[2], Quinn E. Barber[3], George Kourakos[1], and Thomas Harter[1]

[1]Department of Land, Air, and Water Resources, University of California, Davis, USA.
[2]Center for Watershed Sciences, University of California, Davis, USA.
[3]Department of Renewable Resources, University of Alberta, Edmonton, Canada.

*Correspondence to:* Thomas Harter (thharter@ucdavis.edu)

**Abstract.** This study is focused on nitrogen loading from a wide variety of crop and landuse types in the Central Valley, California, USA, an intensively farmed region with high agricultural crop diversity. Nitrogen loading rates for several crop types have been measured based on field scale experiments and recent research has calculated nitrogen loading rates for crops throughout the Central Valley based on a mass balance approach. However, research is lacking to infer nitrogen loading rates

for the broad diversity of crop and landuse types directly from groundwater nitrate measurements. Relating groundwater nitrate measurements to specific crops must account for the uncertainty about and multiplicity in contributing crops (and other landuses) to individual well measurements, and also for the variability of nitrogen loading within farms and from farm to farm for the same crop. In this study, we developed a Bayesian regression model that allowed us to estimate crop or other landuse-specific groundwater nitrogen loading rate probability distributions for 15 crop and landuse groups based on a database of

recent nitrate measurements from 2149 private wells in the Central Valley. The water & natural, rice, and alfalfa & pasture groups had the lowest median estimated nitrogen loading rates, each with a median estimate below 5 kg N ha$^{-1}$ yr$^{-1}$. Confined animal feeding operations (dairies) and citrus & subtropical crops had the greatest median estimated nitrogen loading rates at approximately 269 and 65 kg N ha$^{-1}$ yr$^{-1}$, respectively. In general, our probability based estimates compare favorably with previous direct measurements and with mass balance based estimates of nitrogen loading. Nitrogen mass balance based estimates

are larger than our groundwater nitrate derived estimates for manured forage crops, nuts, cotton, tree fruit, and rice crops. These discrepancies are thought to be due to groundwater age mixing, dilution from infiltrating river water, or denitrification between the time when nitrogen leaves the root zone (point of reference for mass balance derived loading) and the time and location of groundwater measurement.

## 1  Introduction

Nitrate contamination of groundwater is a common problem in agricultural regions across the globe that has also gained increased regulatory attention in recent years. The European Union Nitrate directive, which strives to protect groundwater quality across Europe, reported that as of 2010 all 27 member states had developed action programs to cut nitrogen pollution.



These action programs include monitoring networks, nitrogen application limits, and new technologies for nutrient processing (European Commission, 2010). In California, USA, the Irrigated Lands Regulatory Program (ILRP) was created in 2003 to regulate agricultural water discharge to surface water. In 2012, the ILRP was updated to issue permits for discharge to groundwater and all commercial agriculture is now regulated under the program (California Central Valley Water Board,
5   2016).

Several studies have documented the presence of nitrate contamination in shallow groundwater of the Central Valley aquifer system (Burow et al., 2012, 1998b; Boyle et al., 2012; Lockhart et al., 2013; California State Water Resources Control Board, 2010), where this study is focused. Many people in the Central Valley rely on shallow private wells for domestic use. Studies estimate the number of private wells in the Central Valley to be on the order of 100,000 to 150,000 (Viers et al., 2012; Johnson
and Belitz, 2015). The federal drinking water standard to protect against methemoglobinemia (blue-baby syndrome) is 10 mg/L $NO_3$-N. Background nitrate concentrations are typically less than 2 mg/L $NO_3$-N (Mueller and Helsel, 1996). Drinking water with nitrate concentrations above the background level, but below the drinking water standard, has also been linked to an increased risk of ovarian cancer (Inoue-Choi et al., 2015), thyroid cancer (Ward et al., 2010), bladder cancer (Weyer et al., 2001), and non-Hodgkin's lymphoma (Ward et al., 1996).

Nitrate contamination of groundwater may originate from several sources including synthetic fertilizer, manure, septic systems, and leaky sewer lines. In the Central Valley, California, the major contributors to nitrate contamination are fertilizers and manure applied to crops (Rosenstock et al., 2013; Harter and Lund, 2012). Knowledge of highest-risk crops can aid future regulatory efforts and help in defining priority areas on a county or smaller scale. Previous field-based, often plot-scale research, conducted in California, has measured the amount of nitrogen leached in kg ha$^{-1}$ yr$^{-1}$ from several different crop types includ-
ing grains, vegetables and berries, citrus, nuts, and field crops (Devitt et al., 1976; Embleton et al., 1979; Letey et al., 1977; Pratt et al., 1972; Pratt and Adriano, 1973; Adriano et al., 1972; Allaire-Leung et al., 2001; Liang et al., 2014). However, these studies were largely conducted in the 1970s and 1980s, with little research conducted since then, and are subject to high variability due to various measurement methods (Viers et al., 2012). For example, the historical measurements of nitrogen loading for vegetable and berry crops range from about 20 kg N ha$^{-1}$ yr$^{-1}$ to over 900 kg ha$^{-1}$ yr$^{-1}$ (Pratt and Adriano, 1973; Adriano
et al., 1972; Allaire-Leung et al., 2001; Letey et al., 1977). Two studies estimate nitrogen loading rates from several crop or landuse groups in the Central Valley based on a field-scale nitrogen mass balance approach (Viers et al., 2012; Rosenstock et al., 2013). Loading rates for Central Valley dairy corrals, lagoons, and forage crops receiving manure applications have been estimated based on groundwater monitoring wells located on dairies (Harter et al., 2002; van der Schans et al., 2009). What is lacking in the literature is a regional scale assessment of nitrogen loading to groundwater based on measured groundwater
nitrate data.

The main objective of this paper was to investigate what information may be obtained from existing groundwater quality data that could reveal nitrate loading rates from the large diversity of crop types within the Central Valley. More than 250 crops are grown within the Central Valley alone. By developing an innovative statistical framework, we estimated the probability distributions of crop and other landuse-specific nitrogen loading rates (kg N ha$^{-1}$ yr$^{-1}$) to groundwater from well nitrate data
and historic crop and landuse information around each well. Nitrate loading concentrations vary within specific crop or landuse



types due to differences in farming practices and interactions between hydrogeologic parameters and those practices (farm to farm variation). Loading rates also vary across an individual farm due to small-scale hydrologic heterogeneity (within farm variation). We therefore expect a range of loading rates to apply to any given crop or landuse type. Furthermore, significant uncertainty exists about the source area and the resulting landuses that contributed to a well's nitrate concentration.

Bayesian statistical models have specific benefits when it comes to dealing with uncertainty and complex interactions in groundwater systems: they can incorporate prior knowledge, take into account uncertainty, and allow for variability in predictions. Bayesian methods have been used to estimate nitrate export coefficients from diffuse sources in Swiss watersheds (Zobrist and Reichert, 2006) and for nitrate source apportionment in surface and groundwater (Yang et al., 2013; Ransom et al., 2016; Gaouzi et al., 2013; Xue et al., 2012; Korth et al., 2014). We are not aware of any study that has employed Bayesian methods

to estimate nitrate loading rates to groundwater. We therefore develop a generalized linear model using Bayesian methods and estimate a probability distribution of nitrate loading for crop or landuse types. In contrast to prior studies, estimates here were not based on agronomic data, but on a comprehensive groundwater quality dataset and a geospatial analysis of crops or other landuses surrounding each well. We compiled two large datasets: a database of nitrate measurements from private wells distributed throughout the Central Valley for use in the model and a crop and landuse analysis in the most likely region to have

affected each individual well's nitrate concentration.

## 2   Project area

The California Central Valley (CV) is a large asymmetric, alluvial basin, with the trough axis trending north-west to south-east. The CV is 400 miles long (extending from Red Bluff to Bakersfield, CA), an average of 50 miles wide, and has an area of approximately 20,000 square miles. The boundaries of the CV are the Cascade Range to the north, the Coast Ranges to the west,

the Sierra Nevada mountains to the east, and the Tehachapi Mountains to the south. The CV consists of two separate valleys, divided at the Sacramento-San Joaquin Delta: the Sacramento Valley to the north (northern one-third) and the San Joaquin Valley to the south (southern two-thirds). The CV is filled with six to ten miles of marine and continental deposits. Surface geomorphology consists of overflow lands and lake bottoms, river floodplains and channels, low alluvial plains and fans, and dissected uplands. Post-Eocene continental deposits consist of fine to coarse sediments and compose the major aquifer in the

CV (Page, 1986). Spring 2011 depth to groundwater ranged from 10 feet below ground surface (bgs) in the northern section of the CV to 670 feet bgs in the southern portion of the CV (DWR, 2011).

The CV is a highly productive agricultural region with approximately 7 million of California's nearly 9 million acres of irrigated farmland (California Department of Water Resources, 2010). Major crops grown in the CV are corn, grain and hay, oranges, almonds, peaches and nectarines, cotton, and wine and table grapes. A combination of surface water and groundwater

is used for irrigation. In addition, over 80% of California's 1.8 million adult cows live on dairies in the CV. Total human population for the 19 counties associated with the CV was approximately 7 million in 2014. Major CV cities with over 200,000 residences are Sacramento, Fresno, Bakersfield, Stockton, and Modesto (United States Census Bureau, 2010). Residences located in rural unincorporated areas, many of them clustered in semi-urban belts around smaller towns and major cities, rely





on shallow private wells for drinking and household purposes. Private wells are not regulated in California and it is difficult to know how many may be contaminated by nitrate: a significant portion of the CV has been estimated to contain shallow groundwater with elevated nitrate concentration (Nolan et al., 2014; Lockhart et al., 2013; Ransom et al., 2017).

## 3 Methods

### 3.1 Conceptual model

To use well nitrate measurements for estimating nitrate (as nitrogen) losses to groundwater from specific landuses, we employ a stochastic Bayesian inference model that we derive from physically-based concepts about groundwater dynamics near a pumping well, the location of its source area, and the overlying crop or landuse type. Each crop and landuse type is characterized by water mass flux (recharge rate) and an associated nitrate mass flux to groundwater.

Domestic wells typically supply a single household. The average per household water consumption in California is $1.2 \times 10^3$ $m^3$ $yr^{-1}$ (1 acft $yr^{-1}$, 2 L $min^{-1}$, 0.5 gpm). At a recharge rate of 0.3 m $yr^{-1}$ (1 ft $yr^{-1}$), for example, the source area is approximately 0.4 ha (1 acre). In productive aquifer systems such as that of the CV, this source area typically has a long but very narrow shape (Horn and Harter, 2009). At any time, the water produced by the well is a mixture of water retrieved from a continuous range of depth along the screen of the well representing contributions from across the source area:

$$Q_i(t) = \int\limits_x \int\limits_y q[x, y, t - \tau_i(x, y, t)] dx dy \tag{1}$$

where $Q_i(t)$ is the pumping rate at well $i$ at time $t$, $q()$ is the recharge rate at location $x, y$ at time $(t - \tau)$ (vertical length per time), $\tau_i(x, y, t)$ is the age of the water contributing from point $x, y$ at time $t$. A well's nitrate concentration is therefore also a mixture of nitrate contributions across the source area:

$$C_i(t) = \int\limits_x \int\limits_y \frac{m[x, y, t - \tau_i(x, y, t)]}{q[x, y, t - \tau_i(x, y, t)]} dx dy \tag{2}$$

where $C_i(t)$ is the measured nitrate concentration in well $i$ at time $t$, and $m()$ is the nitrate mass flux associated with the recharge $q()$. The nitrate mass flux is defined by:

$$m[x, y, t - \tau_i(x, y, t)] = c[x, y, t - \tau_i(x, y, t)] * q[x, y, t - \tau_i(x, y, t)] \tag{3}$$

where c is the nitrate concentration in recharge. Note that the above equation implicitly accounts for variability of water and nitrate flux across the surface of the screen.

Under ideal conditions, if $m(x, y, t - \tau)$ is a constant but unknown quantity for a crop or other mappable landuse type, if the source area is known, and if the recharge rate $q(x, y, t - \tau)$ is known, then measurements of $Q_i(t)$ and $C_i(t)$ at N wells is



sufficient to compute $m(x,y,t-\tau)$ and, hence, $c(x,y,t-\tau)$. Writing Equation 2 for each of N wells, where N is the number of crops and landuse types, yields N equations with N unknowns that could be solved exactly.

In practice, neither of these quantities is well known. Despite the availability of regional groundwater flow maps (DWR, 2011) and models (Faunt, 2009), uncertainty about the location of the source area arises from hydrological heterogeneity and largely unknown spatio-temporal variability in large scale groundwater pumping near sampled wells. These factors can greatly alter groundwater flow and thus the source area of a well, often seasonally. Previous studies have therefore used a circular "buffer" zone around each well as an approximation of the well source area (Burow et al., 1998a; McLay et al., 2001; Kolpin, 1996; Lockhart et al., 2013). Circular well buffers have been shown to be reasonable approximations of the potential well source area when the actual contributing source area is unknown (Johnson and Belitz, 2009).

The exact contributions $q(x,y,t-\tau)$ and $m(x,y,t-\tau)$ are also not known. While the crop or landuse type at location $x,y$ is here thought to have a major influence on the magnitude of $m(x,y,t-\tau)$, the specific loading at any location associated with a specific crop or landuse type exhibits within-crop spatio-temporal variability, which arise from variation in farming practices (farm to farm variation) and from hydrogeologic, pedologic, and agricultural practice variability (within farm variation).

When linking well nitrate to nitrate (as nitrogen) mass flux from specific crops and other landuse types, there are therefore three sources of uncertainty: First, the actual source area is but a small sliver of the buffer region. Hence, the contributing sources are uncertain, but constrained by the crop and other landuse type composition within the buffer. Second, the total contribution to mass loading from each crop within the source area may vary due to various factors related to the spatio-temporal variability of agricultural and environmental conditions within the crop area contributing to $Q_i(t)$. Third, the recharge rate $q(x,y,t-\tau)$ is unknown.

The following sections explain the sources of well nitrate data, the computation of the buffer radius, and the mapping of crop and other landuse types that we employed here to demonstrate the usefulness of this approach (Section 3.2). We then explain the Bayesian inference methodology that we employed to account for the aforementioned uncertainties (Section 3.3).

## 3.2 Well and landuse data

A database of nitrate measurements from private wells located in California was compiled from several data sources. The California Ambient Spatio-Temporal Information on Nitrate in Groundwater (CASTING) includes nitrate measurements from private supply, public supply, irrigation, and monitoring wells (Boyle et al., 2012). We selected all well measurements from the CASTING database from supply wells (not monitoring wells) designated as private. We selected samples from private wells because they are typically more shallow than public supply or irrigation wells and are not purposefully located near sources of contamination as are monitoring wells. Private well samples within the CASTING database originated from several sources including the Central Valley Regional Water Quality Control Board (CVRWQCB) Fresno Office dairy domestic wells moni-toring data (sampled for nitrate as a part of the Dairy General Order regulations for dairy facilities in the CV), the California Department of Pesticide Regulation (CDPR), Fresno County, the United States Geological Survey (USGS), Tulare County Environmental Health (TCEH), and the State Water Resources Control Board (SWRCB) Groundwater Ambient Monitoring Assessment (GAMA) Domestic Wells Project in Tulare County. We expanded the original database, which was geographically



limited to the southern CV, to include the data from the same data sources for the entire CV. Also, additional private well samples from the following data sources were added to the CASTING database:

- a set of private wells previously sampled for nitrate as a part of the "Proposition 50 Long Term Risk of Groundwater and Drinking Water Degradation from Dairies and Other Nonpoint Sources in the San Joaquin Valley", funded by the State Water Resources Control Board (SWRCB) (Lockhart et al., 2013) (200 wells total, sampled between 2010-2011),

- additional SWRCB GAMA private wells for Tehama, El Dorado, and Yuba county project areas (GAMA Domestic Well Project, http://www.waterboards.ca.gov/gama/domestic_well.shtml) downloaded from the GeoTracker GAMA online database (http://www.waterboards.ca.gov/gama/geotracker_gama.shtml), and

- CVRWQCB Rancho Cordova Office dairy domestic wells monitoring data provided by the CVRWQCB office.

Records in the database collected between the years 2000 to 2015 were selected. Locations with data collected in multiple years were assigned the median nitrate value of all the recorded measurements in order to prevent multiple samples of the same well and associated landuse. Prior to median aggregation, non-detect nitrate values were replaced with the detection limit and zero values were replaced with the most common detection limit of 2.21 mg/L $NO_3$-$NO_3$. All nitrate measurements were then converted to $NO_3$-N. When geographic coordinates (latitude and longitude) of the private wells in the dairy monitoring program were not available, the wells were located using the dairies street address and placed at the centroid of a dairy's land parcels. The methods for locating the wells varied for each of the other data sources including geographic coordinates, geocoded addresses, offsets by a random small distance, United States Public Land Survey System (PLSS) section, and Assessor's Parcel Number (APN) (Table 1). Due to the well location methods, many wells had overlapping locations. Where multiple wells were geolocated to a single location, a single well was chosen at random to represent that location. Wells outside of the alluvial aquifer system boundary were excluded from the analysis. The final nitrate database had a total of 2149 wells.

Intrinsic aquifer properties were evaluated as an indicator for additional risk for or protection from nitrogen contamination. Here we choose a simple binary indicator: California Department of Pesticide Regulations (CDPR) Groundwater Protection Areas (GWPAs) are 1.6 km square sections that are vulnerable to the leaching of pesticides and are defined by the following criteria: previous detections of pesticides in that section, contains coarse soils and a depth to groundwater less than 21 m, or contains runoff-prone soils and depth to groundwater less than 21 m (California Department of Pesticide Regulation, 2017a). These zones are either vulnerable to contamination due to non-point source leaching of irrigation water "leaching GWPAs" or direct flow paths through hardpan soils (ditches, dry wells, poorly sealed wells, "runoff GWPAs") (California Department of Pesticide Regulation, 2017b). The properties that lead to vulnerabilities from pesticide contamination - shallow depth to groundwater, short residence time in the vadose zone, low reactivity of aquifer sediments - also increase the possibility of nitrate contamination. Wells located within a GWPA zone were attributed as being an indicator for increased risk of nitrate contamination. The non-parametric Kruskal-Wallis statistical test was performed on the nitrate values for wells in each of the two groups (GWPS versus non-GWPA wells). The Kruskal-Wallis test is a ranked one-way analysis of variance which tests whether two groups of values should be considered independent or from the same distribution.



Landuse surrounding wells was analyzed using the California Augmented Multi-Source Landuse (CAML) 50 m resolution raster image file for the year 1990 (Hollander, 2013). CAML was developed from various data sources delineating various natural vegetation types, farmland, and urban areas for five periods of five years each centered on 1945, 1960, 1975, 1990, and 2005. The data sources included the California Department of Conservation Farmland Mapping and Monitoring Program (FMMP), the United States Geological Survey (USGS) National Land Cover Dataset (NLCD) (1992), the California Depart-

ment of Forestry and Fire Protection Fire and Resource Assessment Program Multisource Landcover Layers (MSLC), and the California Department of Water Resources (DWR) Land Use Survey. Importantly, CAML identifies 58 different crop types mapped by the California Department of Water Resources once or twice per decade in each county. Digital maps of these crop types are not available for historic conditions prior to 1990 except through back simulation (Harter et al., 2017). We selected the 1990 rather than the 2010 CAML map to account for some of the time difference between nitrate leached from landuse

practices and the time of groundwater sampling (nitrate travel time) (Ransom et al., 2016). The nearly 60 agricultural and many non-agricultural landuse categories were aggregated into the following land type groups: Water & Natural, Citrus & Subtropical, Tree Fruit, Nuts, Cotton, Field crops, Forage Crops, Rice, Alfalfa & Pasture, Confined Animal Feeding Operation (CAFO), Vegetables & Berries, Peri-Urban, Grapes (including wine and table), and Urban. The Forage Crop group was further separated into fields likely receiving liquid manure irrigation and fields not likely to receive liquid manure based on proximity

to CAFO landuse (within 1.6 km of dairy corrals, lagoons, or facility barns). This analysis does not take into consideration dry manure that may be exported off dairies and applied to crops. Our final study design had a total of 15 landuse and crop groups (hereby referred to as scenario 1). Alternatively, we also analyzed a scenario where CAFO landuse is grouped with (not distinguished from) Manured Forage Crops (14 landuse and crop groups, hereby referred to as scenario 2). The approach presented here can easily be modified to other landuse or management practice categorizations.

Assuming that a private well is very low flow, is screened along its entire depth below the water table, and only intercepts passing water, source area length can be calculated by multiplying well depth by the ratio of specific discharge to groundwater recharge. Landuse amounts (in m$^2$) were quantified within a circular well "buffer" of radius 2.4 km surrounding each well and then converted to a percent of buffer area. The 2.4 km buffer radius was determined by assuming a fixed vertical groundwater recharge rate of 0.30 m yr$^{-1}$, effective horizontal hydraulic conductivity of 30.5 m/day, and hydraulic gradient of 0.001 and

then by the use of Darcy's Law to find specific discharge (calculation details available in Lockhart et al., 2013). The assumed conditions for groundwater recharge, hydraulic conductivity, and gradient are considered representative of the CV (Harter et al., 2002; Horn and Harter, 2009; Phillips et al., 2007).

Groundwater recharge rates can also be variable and were not assumed to be fixed for the purposes of estimating the nitrogen loading rates. Probable vertical groundwater recharge rates (m yr$^{-1}$) were estimated based on results of the Central Valley

Hydrologic Model (CVHM) (Faunt, 2009). The CVHM is a numerical computer model that simulates monthly surface and groundwater flow components throughout the CV for a period of over 40 years. For the purposes of the CVHM, the CV was spatially divided into 20,000 model grid cells (1.6 km$^2$ each) and 10 depth layers (to a depth below ground surface of approximately 550 m). The CVHM further divides the CV into nine textural regions based on estimated aquifer texture. Temporal discretization of the CVHM consisted of 12 monthly stress periods beginning in October 1961 and ending October




2003. For the purposes of our study, flow (m$^3$ day$^{-1}$) below the bottom of the upper model layer (50 ft deep across the majority of the CV) for each monthly stress period occurring in the 1990 decade, was averaged by textural region: we calculated total yearly average flow below the upper CVHM model layer for each CVHM texture region and each year in the 1990 decade

as an estimate of vertical groundwater recharge per year. We used CVHM 1990 model outputs in order to remain consistent with the selected landuse time period. This gave 90 estimates of probable groundwater recharge rates (for each of nine textural zones and each of ten years).

### 3.3 Statistical methods

Bayesian analysis was chosen here as it allows for the estimation of the entire probability distribution of landuse-specific nitrate

leaching concentrations rather than a deterministic value only. Probability distributions for loading rates for the landuse or crop groups described in Section 3.2 were estimated with an exponential distribution generalized linear model using Bayesian methods. Following the structure of the deterministic model (2) and lumping the uncertainties about source area, loading rates, and recharge rates into two stochastic variables representing loading rates and recharge rates only, the basic model equation, modified from Zobrist et al. (2006) is given by:

$$C_i = \frac{1}{\lambda_i} = \sum_{j=1}^{n} \beta_j A_{ij} * (0.1/r) * (I_i + (1 - I_i) * k) \qquad (4)$$

where $C_i$ is the expected nitrate value for well $i$ ( mg/L NO$_3$-N), $\lambda_i$ is the parameter of the exponential distribution, n is the number of landuse categories considered, $\beta_j$ is the unknown nitrogen loading rate from landuse $j$ (in kg N ha$^{-1}$ yr$^{-1}$), $A_{ij}$ is the percent of well buffer $i$ that is landuse $j$, 0.1 is a conversion factor to convert units of mass to units of concentration based on vertical groundwater recharge rate (with units [m*(mg/L)/(kg ha$^{-1}$]) (Pratt et al., 1972), $r$ is the vertical groundwater

recharge rate (in m yr$^{-1}$) , $I_i$ is an indicator variable representing whether or not well $i$ is located within a GWPA (0 for outside and 1 for within), and $k$ is a groundwater protection parameter representing a mean decrease in nitrate values applied only to wells outside GWPA zones. We assume nitrate is the dominant form of nitrogen within and persisting in the saturated zone (Liao et al., 2012). The percent of each landuse surrounding each well was calculated as described above. Note that Equation 4 represents a general model that can be applied to an arbitrary number of crop and other landuse type classes, source area

configuration, recharge rate distribution, and indicator variables, depending on the application.

Bayesian analysis requires the assumption of an initial probability distribution for each parameter to be estimated and this represents the current estimate or knowledge of the parameter. The initial probability distribution assigned to each parameter to be estimated is known as a prior probability distribution or "prior". Priors are updated in the modeling process and the final estimate is known as a "posterior" probability distribution.

Student t-distribution priors, representing an initial approximation of the potential nitrogen loading rate, were assumed for $\beta_j$. For an un-biased assessment, each of the landuse or crop groups were given the same t-distribution prior which represents the potential and unknown loading rate. We used the form of the t-distribution parameterized by the location, scale, and degrees of freedom (Plummer, 2015a). The location parameter of the common t-distribution prior was set to equal the median measured




nitrate value of approximately 5 mg/L (see Results). Reasonable, non-informative degrees of freedom and scale are 1 and 25, respectively. The choice of t-distribution parameters reflects a heavy-tailed/high variance distribution which gives the model flexibility to move the final predicted loading rates (posterior distributions) away from the prior distribution based on evidence observed in the measured nitrate values and surrounding landuse. The t-distribution priors were truncated at zero in order to

ensure that the estimated concentrations cannot be negative.

A non-informative Student-t prior probability distribution was used as an initial approximation of the GWPA factor $k$. Appropriate location, scale, and degrees of freedom were 0.5, 1, and 1, respectively (see Results section). A log-normal distribution was used for the prior probability distribution of potential recharge rates with location and scale of -2 and 0.6, respectively (see Results section). The prior probability distribution for the GWPA factor was truncated at zero to eliminate any potential for

negative values. The recharge rate, $r$ was assumed to be positive.

Markov Chain Monte Carlo (MCMC) methods were used to infer the marginal posterior distributions of nitrogen loading rates for each landuse. MCMC was performed using the Gibbs sampler JAGS (Plummer, 2003). JAGS was run from within the statistical computing program, R (R Core Team, 2016), using the package rjags (Plummer, 2015b). Two chains were run and the first 100,000 realizations of each chain were discarded as the burn-in period. After burn-in, each chain was sampled 200,000

times with a thinning interval of 400 to reduce autocorrelation. Traceplots were visually inspected to confirm convergence and proper mixing of chains, after which the realizations from each chain were combined for a total of 1,000 realizations per parameter. 95% and 68% credibility intervals were then calculated for each parameter. Final model run time was approximately 12 hours on an Intel Core I-5 4670 processor with 16 GB of 1600 MHz DDR3 RAM.

We calculated a standardized Pearson goodness-of-fit statistic (McCullagh and Nelder, 1989) for each scenario. The goodness-

of-fit statistic used the average of the raw residuals between the measured nitrate value for each well and the estimated value (realization) divided by the standard deviation of all of the realizations for each well (1000). The sum of the squared average raw standardized residuals was then divided by the total number of wells minus the number of parameters in the model (17 for scenario 1 and 16 for scenario 2) to give the goodness-of-fit.

## 4  Results

The wells in the database compiled for this study had a minimum nitrate concentration of non-detect, median of 4.95, and maximum of 131.0 mg/L (Figure 1). Median nitrate value for wells located within GWPAs was more than twice the median nitrate value for wells located outside GWPAs (8.22 mg/L $NO_3$-N versus 3.92 mg/L). The non-parametric Kruskal-Wallis test for independence between the two groups of well's nitrate values was significant at the 95% confidence level. The fact that GWPA zone wells had higher nitrate than non-GWPA zone wells justifies the use of a GWPA related groundwater protection

term in the model (Equation 4). Median nitrate value was calculated for runoff versus leaching GWPAs and the values were within 1 mg/L, therefore runoff and leaching GWPA zones were grouped for the context of this study (Section 3.2). The model estimated posterior distribution for the non-GWPA protection factor, $k$, was fairly narrow, with a 95% credibility interval (CI) of 0.699 to 0.850 and a median of 0.773 (Figure 2).





Median depth to top and bottom of well screen for all wells with depth information available (915 wells) was 40.5 m (133 ft) and 64 m (210 ft), respectively. Minimum and maximum depth to top of well screen was 2 m (6 ft) and 184.5 m (602 ft), respectively and minimum and maximum depth to bottom of well screen was 6 m (20 ft) and 274.5 m (900 ft).

Estimated groundwater recharge from the CVHM model used as the basis for the recharge rate ($r$, Equation 4) parameter was between 0.016 and 0.530, with a median of 0.146 (m year$^{-1}$) (Figure 3). Our Bayesian model estimated (updated, posterior) recharge rate was slightly greater (posterior 95% credibility interval of 0.159 to 0.481 and median of 0.281 m year$^{-1}$ for scenario 1), but still within the range of the CVHM model estimates (Figure 3).

Nitrate concentrations for each well were compared to the percent of each landuse or crop group within well buffers for scenario 1 with a locally weighted scatterplot smoothing (lowess) line for each plot (Figure 4). A lowess line is a smoothed regression line that represents many locally weighted polynomial fits to the data by weighted least squares (Cleveland, 1979). As Citrus & Subtropical crops, Manured Forage crops, and CAFO landuse proportions increased, nitrate in well samples appeared to increase (these groups were expected to have greater predicted nitrogen loading rates).

The standardized Pearson goodness-of-fit statistic, a summary measure of squared deviations between observations and their estimated values, has a value of near 1 for good-fitting models (McCullagh and Nelder, 1989). The standardized Pearson goodness-of-fit statistics were 1.16 and 1.17, respectively for scenario 1 and 2 and therefore the model was understood to fit the data fairly well.

Estimated nitrogen loading rates across all crop and landuse groups for both scenarios ranged from between negligibly small to nearly 600 kg N ha$^{-1}$ yr$^{-1}$ (Figure 5 and Table 2). The scenario 1 CAFO group and the scenario 2 Manured Forage & CAFO group had the greatest estimated nitrogen loading rates, while Alfalfa & Pasture, Rice, and Water & Natural for both scenario 1 and 2 were the lowest (Figure 5 and Table 2). The scenario 1 CAFO group also had the greatest range of estimates, reflecting the highest degree of uncertainty or spatial variability. Results for both scenario 1 and 2 were fairly consistent, with the exception of the scenario 1 CAFO and scenario 2 Manured Forage & CAFO groups. When Manured Forage, which occupies relatively large areas, was grouped with nitrogen intensive, but small area CAFO landuse in scenario 2, the estimate for Manured Forage & CAFO was over three times lower than the estimated nitrogen loading for CAFO alone. Similarly, the estimate for Manured Forage & CAFO in scenario 2 was about two times higher than for Manured Forage alone in scenario 1. The effect of merging the two groups in scenario 2 yielded only small changes in the estimates for the other crops and landuses; several estimated loading rates increased slightly, such as for Vegetable & Berry crops, while others decreased slightly (Field Crops), or remained approximately the same.

CAML landuse groups for scenario 1 are plotted side by side with corresponding Bayesian model estimates median nitrogen loading rate in order to spatially represent relative risk to groundwater from nitrate contamination (Figure 6). Low estimated nitrogen loading is concentrated in the northern CV, where Water & Natural landuse and Rice crops are dominant, along the middle eastern and southern eastern and western edge where Water & Natural landuse is dominant, or scattered along the central axis of the CV following the pattern of Alfalfa & Pasture crops. The greatest nitrogen loading is scattered randomly throughout the central CV from north to south (a direct representation of CAFO locations), or echoes the pattern of Citrus & Subtropical crops along the eastern edge of the southern CV in Tulare and Kern counties.



## 5 Discussion

5  The Bayesian estimation model provides a probabilistic data-driven evaluation of nitrogen loading rates to groundwater from various landuses. Unlike previous estimates, the model here represents an inverse model estimate of nitrogen loading to groundwater using nitrate concentrations from a large number of groundwater production wells. Given the relative proportion of landuse groups within the source area and an estimate of recharge rates, groundwater concentrations are transformed to effective nitrogen loading rate distributions for each landuse group.

10  To determine whether the Bayesian model yields realistic estimates, we compare model results to two alternative, mutually independent datasets of nitrogen loading to groundwater: field measurements of nitrogen loading, obtained using a variety of field-based measurement techniques, and potential groundwater nitrogen loading obtained by closure to nitrogen mass balance based estimates of historic nitrogen fluxes in the CV. For further evaluation of the Bayesian loading estimates, we also consider hydrologic conditions other than mass loading that may affect nitrate concentrations measured in wells and used for 15  the Bayesian loading estimation.

Studies determining nitrogen loading to groundwater from agricultural crops have historically used soil samples, anion exchange resin bags, suction lysimeters, or tile drain samples (Devitt et al., 1976; Embleton et al., 1979; Letey et al., 1977; Pratt et al., 1972; Pratt and Adriano, 1973; Adriano et al., 1972; Allaire-Leung et al., 2001; Liang et al., 2014). These field measurements are limited to a few crops and are not available for all crop groups analyzed in this study. Measurements were 20  available for 5 crop groups: Citrus & Subtropical, Vegetables & Berries, Cotton, Alfalfa & Pasture, and Rice. Our Bayesian model results for these crop groups were generally consistent with the field measurements of nitrogen loading (Figure 7). For each crop group, multiple field measurements overlap with the 95% credibility interval (CI) of the Bayesian nitrogen loading estimates. For Rice, all three available field measurements overlap with the Bayesian loading model estimates. Overall, the field measurements encompass a wider range of values and extend to larger values than the 95% CI of our model estimates, 25  especially for Vegetables & Berries.

The high variability of nitrogen loading rates measured (field measurements) within crop groups, especially for Citrus & Subtropical, Vegetables & Berries, and Cotton (Figure 7) is the result of several factors including within field crop rotation, variable irrigation and farming nutrient management practices within fields and among farms, and variable measurement methods. The field nitrogen loading measurements therefore need to be interpreted with some caution (Viers et al., 2012). The 30  Bayesian loading estimates appear to confirm many of the field measurements, given the overlap of measured with estimated distribution of nitrogen loading. The range of loading rates predicted by the Bayesian model may therefore be interpreted as representing both, the potential variability of loading rates within a landuse group, and uncertainty about the loading rate.

Detailed spatially and temporally distributed nitrogen flux analysis for the Central Valley has been performed and documented for the Central Valley and Salinas Valley Groundwater Nitrogen Loading Model (GNLM) (Viers et al., 2012; Rosen- 35  stock et al., 2013; Harter et al., 2017). Briefly, the conceptual basis for the GNLM is a mass balance analysis of nitrogen fluxes into and out of agricultural crops, at the field scale, including nitrogen in atmospheric deposition, irrigation water, synthetic fertilizer, manure, wastewater effluent, harvest, runoff, and atmospheric emissions from soils. Potential groundwater nitrogen





loading from agricultural cropland was computed as closure to the mass balance. GNLM accounts for typical nitrogen fertilizer and harvest rates of 58 individual crops, spatially distributed across the CV. It also considers locally and regionally varying

nitrogen deposition, irrigation water nitrate, and facility specific manure and waste water effluent applications to agricultural crops from CAFOs, wastewater treatment plants, and food processors. As a result, GNLM estimates of groundwater nitrogen loading within a crop group can be highly variable due to variability between crops within a group and due to local variability in non-fertilizer nitrogen fluxes. For consistency with the approach used here, we compare 1990 GNLM results to the Bayesian model results.

The Bayesian model results overlap with GNLM results for Citrus & Subtropical, Vegetables & Berries, Field Crops, Grapes, and the Water & Natural group (Figure 8). Median values for these crop groups obtained from the groundwater data (Bayesian analysis) were 39% (Citrus & Subtropicals), 20% (Vegetables & Berries), 10% (Field Crops), and 1% (Grapes) lower than mass balance based estimates (GNLM). In the Bayesian analysis, Citrus & Subtropical and Vegetable & Berries yielded the second, and third largest crop group median rates (scenario 1). The median GNLM rate is somewhat higher, but within the 68%

CI estimated with the Bayesian model for both crop groups. The lower end of the range of GNLM estimates for these two crop groups is lower than predicted by the Bayesian model. The Bayesian model distribution extends to similar high concentrations as GNLM at the upper end of the predicted range for Citrus & Subtropical, but is about 50% lower than the upper end of the GNLM prediction for Vegetables & Berries (Figure 8).

The high loading rates estimated by our model for Citrus & Subtropical and for Vegetables & Berries appear to confirm the

large difference between fertilizer and harvest rates for crops in this crop group. On the other hand, the similarity between our results and GNLM results for groundwater nitrogen loading from Field Crops and Grapes confirms the lower fertilization rates and resulting lower nitrogen surplus typically occurring in these latter crops (when not manured). For Citrus & Subtropical, the estimated high rates may also be a result of the significant potential for direct contamination pathways induced by farming practices thought to be common in the region where these crops are grown, along the eastern edge of the valley floor in Tulare

and Fresno counties (Figure 6). This region includes a significant portion of "runoff" designated GWPA zones on soils that contain a shallow hardpan layer (Troiano et al., 2014) and where dry wells used for surface drainage are common (DeMartinis and Royce, 1990). In addition, the water table in these same regions is relatively shallow (7 – 10 m bgs) (DWR, 2011). Infiltration of agricultural surface runoff through dry wells and/or a shallow depth to water may lead to more rapid nitrogen loading at the high rates predicted by our model for this crop group than for other crops/landuses.

Bayesian model loading rates within the 95% CI of the Alfalfa & Pasture and Water & Natural groups had the lowest overall values (Figure 5 and Table 2). The Bayesian results for these crop groups are driven by the lack of apparent correlation between an increase in nitrate concentration in wells and increasing proportions of their respective area within well buffers (Figure 4). From a nitrogen mass balance perspective, low estimated nitrogen loading rates are expected for both landuse categories because fertilizers and manure are not typically applied to these areas: alfalfa is a legume, which has the ability to

fix atmospheric nitrogen rather than relying on synthetic fertilizer; Water & Natural landuses are only subject to atmospheric nitrogen deposition and some symbiotic nitrogen fixation). In comparison, GNLM assigned a single value for groundwater





nitrogen loading from alfalfa (30 kg N ha$^{-1}$ yr$^{-1}$), based on reported field measurements (Viers et al., 2012). The assigned value in GNLM is much higher and outside the 95% CI estimated with the Bayesian model.

For urban landuse, GNLM assigned 20 kg N ha$^{-1}$ yr$^{-1}$ based on a review of urban nitrogen leaching (Viers et al., 2012), which is within our model estimated 95% CI for Urban nitrogen loading. In the Bayesian model results, Peri-Urban areas have a greater predicted 95% CI when compared to Urban (Figure 5). Peri-Urban areas are defined as rural homesteads. Each well buffer contained Peri-Urban areas. Peri-Urban areas were expected to have a greater nitrogen loading rate than Urban due to the use of septic systems. Septic systems are common outside urban areas not reached by centralized sewer services. Loading from these areas can be highly variable depending on septic system density. Our model estimated 95% CIs for Peri-Urban areas overlaps with the range for nitrogen loading from septic systems obtained by considering the density of households using septic, 10 to over 50 kg N ha$^{-1}$ yr$^{-1}$ (Viers et al., 2012). The Bayesian results for loading rates from Peri-Urban areas are consistent with research indicating that domestic wells in areas with higher septic system density are at significant risk to intercept septic system leachate (Bremer and Harter, 2012).

The CAFO landuse group, like the Citrus & Subtropical crop group, exhibited positive apparent correlation between nitrate concentration in wells and its area fraction in well buffers (Figure 4). In the Bayesian analysis (scenario 1), CAFO had the greatest estimated median loading rate among all crop and landuse groups (269 kg N ha$^{-1}$ yr$^{-1}$, Table 2). The value is about 50% higher than the value used for dairy corrals in the GNLM study (183 kg N ha$^{-1}$ yr$^{-1}$), and about one-quarter of the GNLM value for dairy lagoon loading to groundwater (1171 kg N ha$^{-1}$ yr$^{-1}$), which represents the average loading rate obtained from extensive field monitoring (Luhdorff and Scalmanini Consulting Engineers, 2015). However, both, dairy corrals and lagoons are included in the CAFO category for the Bayesian analysis. For the CV, Harter et al. (2017) estimated the total dairy corral area to be 12,200 ha and the total dairy lagoon area to be nearly 2,400 acres. Hence, the area weighted average loading rate from both areas is about 340 kg N ha$^{-1}$ yr$^{-1}$, well within the 68% CI of our Bayesian estimate. At 565 kg N ha$^{-1}$ yr$^{-1}$, the upper bound of the Bayesian 95% CI estimated for CAFO is much higher than average area-weighted corral and lagoon loading reflecting the large variability in groundwater nitrogen loading from this landuse apparent in groundwater nitrate values. Similarly, large variability has been observed in other research, particularly from dairy lagoons (Ham, 2002; Luhdorff and Scalmanini Consulting Engineers, 2015). VanderSchans et al. (2009) provided estimates specific to two dairies located on well-drained soil with shallow groundwater: 872 and 807 kg N ha$^{-1}$ yr$^{-1}$ for the dairy corral and dairy lagoons at the site, respectively. These estimates are greater than the 95% CI of the Bayesian model estimate, but confirm that the upper end of our estimated CI is not unreasonable.

Crop groups for which the groundwater nitrate based Bayesian model estimates are much lower than mass balance based GNLM estimates include Manured Forage, Nuts, Cotton, Tree Fruit, and Rice. GNLM results for these crops are driven mostly by the difference between applied synthetic fertilizer or manure and harvested nitrogen (Figure 8): For Rice, the Bayesian estimate is less than 5 kg N ha$^{-1}$ yr$^{-1}$, while GNLM predicts residual root zone losses to be 20 kg N ha$^{-1}$ yr$^{-1}$. Bayesian estimates of median loading rates for Cotton, Nuts, and Tree-Fruit range from 12 to 27 kg N ha$^{-1}$ yr$^{-1}$, while the GNLM estimates for these crop groups range between 90 and 110 kg N ha$^{-1}$ yr$^{-1}$. There is no overlap in CIs, between the two method estimates. However, measured field data for Cotton overlap with both method's CIs.





The discrepancy between the Bayesian analysis and other data for Rice, Tree Fruit, Nuts, Cotton, and possibly Manured Forage indicate that other processes, not explicitly accounted for in the Bayesian analysis, potentially attenuate the impacts of nitrogen mass loading, relative to field mass balance based estimates. In the Bayesian method, these processes, discussed below, lead to lower effective loading rate estimates when considering current groundwater quality data.

The distinct distributions obtained with scenario 1 simulations for CAFO and Manured Forage Crop indicates a statistically strong signal differentiating loading from these two landuse groups, despite the fact that the the two are typically located immediately adjacent to one another. CIs for CAFO and Manured Forage did not contain mutually overlapping values. The results correspond to differences found in previous research that estimated loading from manured forage crops to be nearly half of that from lagoons or corrals. VanderSchans et al. (2009, for the same location as the corrals and lagoons above) estimated loading rates of 486 kg N ha$^{-1}$ yr$^{-1}$for Manured Forage crops. In a separate study, shallow groundwater monitoring well nitrate indicated leaching from manured forage fields of 280 kg N ha$^{-1}$ yr$^{-1}$ (Harter et al., 2002). While lower than CAFO estimates in these studies, both estimates are much larger than our Bayesian model estimates for Manured Forage crops in scenario 1, which estimates the upper end of the 95% CI to be less than 100 kg N ha$^{-1}$ yr$^{-1}$. GNLM estimates for manured crops also typically far exceed 200 kg N ha$^{-1}$ yr$^{-1}$. Our Manured Forage estimates may partly be lower due to an overestimation of land area assumed here to be used for manure application (any forage field within 1.6 km of a dairy). Actual manure distribution in 1990 varied and may have taken up much less forage crop area. Non-manure forage may therefore be partially mixed into the category Manured Forage.

Due to Manured Forage crops and CAFO typically being located adjacently, we also considered a scenario 2, where CAFO was lumped with Manured Forage into a single landuse category. The proportion of scenario 1 CAFO landuse within well buffers was small (almost all wells had less than 10% CAFO landuse within the buffer, Figure 4), while Manured Forage crops occupied a larger proportion of the area (up to 50% of well buffers were Manured Forage crops, Figure 4). For CAFO landuse above about 0.05, higher nitrate concentrations were indicated, while higher concentrations were most dominant for Manured Forage landuse above 0.35 (Figure 4). Scenario 2 results represent an effective, area-weighted nitrogen loading across all dairy related landuses: corrals, lagoons, and manured crop areas. In the CV, Manured Crop areas are estimated to take up 174,000 ha, more than a magnitude larger than the CAFO area (corrals and lagoons: 12,200 ha) (Harter et al., 2017). The much larger area of Manured Crops when compared to CAFO explains why the range of the estimated lumped nitrogen loading rate for Manured Crops and CAFO in scenario 2 is much closer to the range of scenario 1 Manured Forage Crops than to scenario 1 CAFO results. The process of merging CAFO landuse with the surrounding Manured Forage landuse reduces the estimated loading rates that would otherwise be specific to CAFO landuse (mostly lagoons and corrals), but may provide a more representative estimate, given the uncertainty about past manure application areas, for CAFO and associated (partially) manured areas as a whole. We note that the similarity of results between scenario 1 and scenario 2 obtained for other crop and landuse groups indicates that the Bayesian method is robust to the particular choice of crop and landuse groupings.

The discrepancy between some mass balance estimates and the Bayesian model estimates may be due to several hydrologic processes that affect well nitrate concentrations independent of nitrogen loading rates in the recharge area of a well. These include dilution with older groundwater, mixing with recharge water from streams, and denitrification or ammonium





volatilization in the vadose zone or in groundwater (Ransom et al., 2017). Dilution of nitrogen in recharge water is most likely to occur through mixing along the well screen with older, low nitrogen containing, water. Mixing with old water (that recharged

prior to the advent of nitrogen fertilizers in the 1930s and 1940s) within well screens could potentially have affected the model estimated loading rates for all crop and landuse groups. Due to the length of well screens, all domestic well samples contain water of mixed age. A study located near Fresno, CA (within our study area) found that groundwater samples from individual wells contained groundwater with an age range typically greater than 50 years (Weissmann et al., 2002). Weissmann et al. (2002) attributed the high variance in groundwater residence time within a single well to the heterogeneity within the alluvial

aquifer system which produced spatially varying flow velocities. Weissmann et al. (2002) also reported significant positive skewness (tailing) in the distribution of groundwater ages within individual wells, meaning wells contained some groundwater which was much older than the median age. The authors reported the tailing behavior was due to low hydraulic conductivity units within the aquifer in which slow advection and diffusion dominate the transport process. These results are similar to an earlier study in the Salinas Valley, California (an alluvial aquifer system that, at comparable depth, is similar to the CV and

dominated by agriculture) where the authors found significant dispersion of groundwater ages within simulated groundwater samples (Fogg et al., 1999). Simulated water samples from Fogg et al. (1999) had groundwater ages ranging from 10 years to greater than 500 years. The authors point out that the water pumped from wells in the Salinas Valley was only partially from water that was young enough to be contaminated by nitrate and that this proportion would only increase in the future.

Geostatistical analysis of groundwater age tracers from wells sampled in the CV has estimated the depth to the top of well

screens pumping pre-modern (age of 60 years of more) groundwater to be between 30 - 120 m (Visser et al., 2016). According to those results and considering the median depth to bottom of well screen for wells in our study (64 m), a portion of our study wells screened intervals likely penetrate the interface between young and old water. Therefore, mixing within wells with water recharged prior to the intensive use of fertilizers, with water with long residence times (tailing effect), or with water recharged between the 1940s and 1970s, when nitrogen losses for many crops were smaller than in 1990 or later (Harter et al., 2017) could

have led to the lower estimates of nitrogen loading for some crops in this study. Mixing with groundwater recharged prior to 1990 may play a significant role in the Bayesian estimates obtained for Manure Forage, Cotton, and Nuts: Significant increases in manure application to forage crops occurred only after the 1960s, with large increases in the 1980s and 1990s (Harter et al., 2017). These changes may not yet have affected much of the water drawn from the measured wells. For Cotton and Nuts, field mass balance based estimates for nitrogen loading indicate much lower median rates in 1975 and 1960 (about 40 kg N ha$^{-1}$

yr$^{-1}$). Also, the harvested area for nut crops increased sharply between 1960 (64,000 ha) and 1990 (250,000 ha) (Harter et al., 2017). At individual wells, or even across our set of wells, it is difficult to further assess the dilution effect with older water without more detailed analysis of groundwater age throughout the CV and more information on study well screened intervals.

Infiltrating river water with very low nitrate concentration is a significant source of recharge in some areas. This may also dilute otherwise high nitrogen concentrations in land surface recharge. Wells near rivers may receive a significant fraction of

river recharge. A study focused on the TLB (within the CV) geospatially related areas near rivers with lower nitrate concentration in wells. The study highlighted areas where major rivers flow into the TLB from the Sierra Nevada Mountains and found that these areas were also characterized by wells with lower relative nitrate concentrations (Boyle et al., 2012). A statistical





analysis of CV groundwater nitrate recently confirmed that proximity to major streams is a significant controlling factor for a wells' nitrate concentration (Ransom et al., 2017).

Boyle et al. (2012) point out that agricultural areas near rivers are also more likely to receive surface water irrigations. Nitrogen loading from fields receiving low nitrate surface water irrigations is likely to be lower than from fields irrigated with nitrate contaminated groundwater (Boyle et al., 2012). Irrigation water source may have impacted our model estimates and resulted in the lower loading rates compared to mass balance estimates for some crops.

Denitrification and ammonium volatilization could also play an important role in some differences between our model
estimates and mass balance estimates, though we do not suspect widespread regional denitrification. A study focused in the San Joaquin Valley correlated anoxic groundwater conditions to lower nitrate concentrations, but the authors did not attribute this to denitrification (Landon et al., 2011). Instead, Landon et al. (2011) attributed the lower nitrate concentrations in wells with water classified as anoxic to older groundwater with longer residence times (recharged prior to the intensive use of fertilizers). Landon et al. (2011) did not find significant decreases in nitrate concentration in wells due to denitrification. Results of a multi-
model averaging approach to estimate oxygen and nitrogen reduction rates in the San Joaquin Valley did estimate denitrification rates to be significant (Green et al., 2016). However, Green et al. (2016) also estimated oxygen reduction rates to be low with a median of 0.12 mg L$^{-1}$ yr$^{-1}$. Much of the shallow groundwater in the CV is well-oxygenated: the dissolved oxygen content of Lockhart et al. (2013) study wells with a measurement (Table 1) was above 5 mg/L on average (Ransom et al., 2016). In addition, Green et al. (2016) found that the estimated rates of oxygen and nitrogen reduction would not protect wells from
nitrate contamination, given current nitrogen application rates. We therefore do not expect that denitrification or ammonium volatilization had a significant, overall effect on our model results, but rather may have had an isolated effect in areas with Rice and possibly with Manured Forage. For example, a study on four rice fields in the Sacramento Valley (northern CV) found little to no nitrate leaching below the rice root zone (pore water nitrate levels were typically below approximately 2.5 mg/L NO$_3$-N). This was attributed to denitrification during the rice growing season when fields are flooded, ammonia volatilization,
plant uptake, and crop management practices that contribute to the development of a hardpan layer directly below the rice root zone. The study also found very low nitrate concentrations in groundwater wells near rice fields (median value less than 1 mg/L NO$_3$-N) (Liang et al., 2014) consistent with our estimates of Rice nitrogen loading. The GNLM mass balance estimates are outside the range of our model predicted CIs for Rice as they reflect nitrogen losses prior to denitrification or ammonium volatilization potentially taking place in saturated clay soils of rice fields.

Denitrification may also explain why the attenuation factor for areas outside GWPA protection areas is significantly lower than 1 (Figure 2). Regions outside GWPAs are characterized by larger depth to groundwater (greater than 21 m). The 15% to 30% lower apparent nitrogen concentrations in the less vulnerable regions may be due to longer travel times in the deep vadose zone or some additional denitrification and ammonium volatilization in the heavier soils or the underlying deep vadose zone, or in groundwater.

Our model estimates a greater median groundwater recharge rate ($r$) compared to the prior information from the CVHM model. This is likely because the study wells are concentrated in agricultural areas with greater recharge rates due to irrigation. The CVHM (Faunt, 2009) estimated recharge rates are calculated for the entire Central Valley, including natural areas with few





wells and little agriculture. Many of our study wells were spatially clustered in the Tulare/Kern and Kings subbasins, which had median CVHM estimated recharge rates of 0.21 and 0.35 (m year$^{-1}$), respectively for the 1990 decade. These median rates are near our model estimated median recharge rate of 0.281 (m year$^{-1}$) (Figure 3).

## 6   Conclusions

The novel Bayesian tool developed here provides a robust statistical methodology to relate nitrate measurements in wells to the
various types of surrounding landuses as a means to obtain a statistical distribution of nitrate loading rates. After accounting for some hydrologic processes not explicitly represented in the approach (denitrification, ammonium volatilization, mixing with older water or water recharge from streams) the Bayesian model estimates were consistent with previous independent estimates and measurements of potential groundwater nitrogen loading. The validation against independently obtained data demonstrate the general usefulness and accuracy of the Bayesian nonpoint source pollutant loading model introduced here. The information
can provide a better assessment of landuse impacts to water quality based on extensive nitrate and other nonpoint source groundwater contaminant data measured in private wells. The tool can be used to define high nitrogen loading (high risk) zones (Figure 6). As is apparent from Figures 1 and 6, much of the CV already suffers from or is at risk for serious groundwater contamination by nitrate. Our results indicate that the highest nitrogen loading rates are associated with Confined Animal Feeding Operations (dairies) and their associated feed crops (with the exception of alfalfa), as well as from Citrus & Subtropical
crops and Vegetable & Berry crops. Yet, interactions between the depth to older water, well construction, direct contamination pathways, groundwater depth, presence of river water recharge and landuse have likely affected the amount of nitrate pumped by wells. Estimates of nitrate loading generally correspond to previous field measurements or mass balance estimates. For Nuts, Cotton, Tree Fruit, and Rice estimated nitrogen loading rates were lower than mass balance estimates. Nuts, Cotton, and Tree Fruit estimates may have been affected by dilution of crop leachate water past the root zone by infiltrating low-nitrate river water
or by mixing with older low-nitrate water within the well screen. Land managers may default to the mass balance estimates for those crops. Rice estimates were likely lower than mass balance estimates due to denitrification and ammonium volatilization directly below saturated rice fields, which mass balance estimates did not consider. Estimates of nitrate leaching concentration for particular crop and landuse types, obtained with this tool may not be generalized and transferred to regions with substantially different climate, agronomic, geologic, geomorphic, or soils conditions. However, the statistical modeling approach provided here is broadly applicable to other semi-arid, irrigated regions underlain by alluvial aquifers and to nonpoint source pollutants other than nitrate, e.g., salinity and pesticides. Our results could potentially be improved with more information on groundwater age and portion of older water pumped by the wells in our study. Also, a potential limitation of the method is the limited
5 availability of historic crop and landuse maps of sufficient resolution, corresponding to typical groundwater age found in wells. In a similar vein, regional scale models capable of simulating physical groundwater flow processes described by Weissmann et al. (2002) and Fogg et al. (1999) are needed to more accurately access the transport of nitrogen in the subsurface.





# 7   Code availability

Model code may be available upon request.

# 8   Data availability

This dataset is available upon request. Please note well location information, well data group, or well names are not publicly

available due to confidentiality agreements with well owners.

*Author contributions.*  Dr. Thomas Harter designed the research question and experimental, conceptual, and overall statistical structure. Katherine M. Ransom, with assistance from Quinn Barber, prepared the initial well data. Andrew M. Bell processed landuse layers for the dataset. Katherine M. Ransom with assistance from Mark N. Grote and Arash Massoudieh wrote and ran model code and assessed model results. George Kourakos and Thomas Harter prepared and processed Groundwater Nitrogen Loading Model (GNLM) results for use in this

study. Katherine M. Ransom prepared the manuscript text and figures.

*Competing interests.*  The authors declare that they have no conflict of interest

*Acknowledgements.*  We are extremely grateful to Mark N. Grote and Arash Massoudieh for their help with model development and assessment, our work would not have been possible without you. Thank you to Jo Ann M. Gronberg for preparing figure map templates of the Central Valley, California and to Claudia C. Faunt and Jon Traum for providing and processing the CVHM data for our initial estimates

of groundwater recharge rates. Partial funding for this work was provided through the State Water Resources Control Board (SWRCB) Agreement No. 09-122-250, and through a grant from the California Department of Agriculture Fertilizer Research and Education Program, project numbers 11-0301 and 15-0454. The authors gratefully acknowledge the financial support of the Center for Watershed Sciences made possible by a gift from the S. D. Bechtel, Jr. Foundation.



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





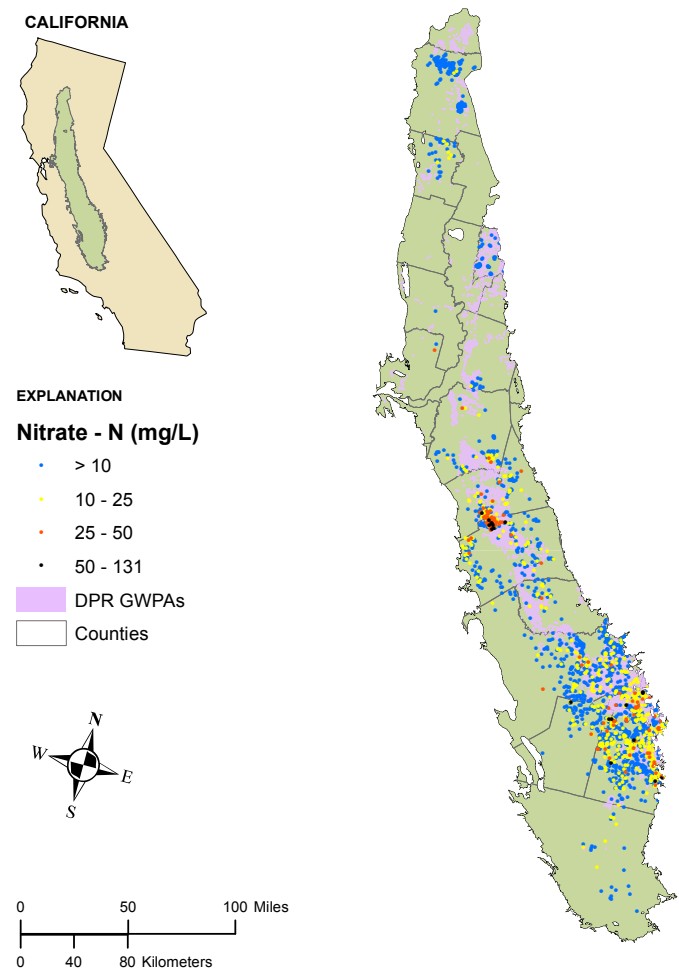

**Figure 1.** Study well locations color coded by nitrate (NO$_3$-N mg/L) value overlain with CDPR GWPA zones (runoff and leaching undifferentiated).





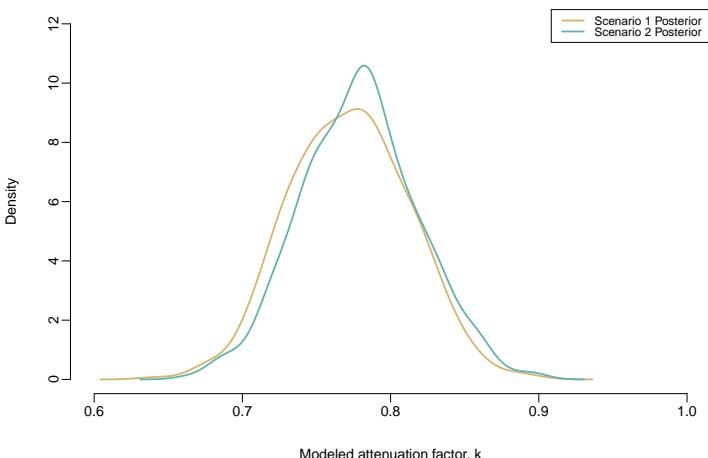

**Figure 2.** Posterior probability density for the non-GWPA attenuation factor, $k$, for scenario 1 (tan) and 2 (teal).

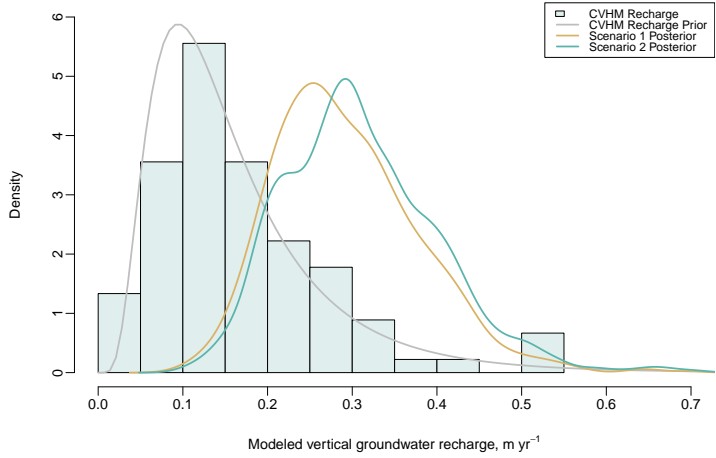

**Figure 3.** CVHM estimated annual vertical groundwater recharge (grey bars), log-normal prior probability density for recharge input to model (grey line) and model estimated posterior probability density for the annual recharge rate for scenario 1 (tan) and 2 (teal).







**Figure 4.** Scatterplot of proportion of landuse within each well buffer versus well nitrate concentration for each of the 15 landuse or crop groups in scenario 1. The red line is a locally weighted scatterplot smoothing line (Cleveland, 1979). Note that each plot shows nitrate concentrations between 0 and 25 mg/L NO$_3$-N (approximately 5 times the median value), however all data was used to calculate the plotted smoothing lines. The x-axis is scaled differently among subplots for better resolution.





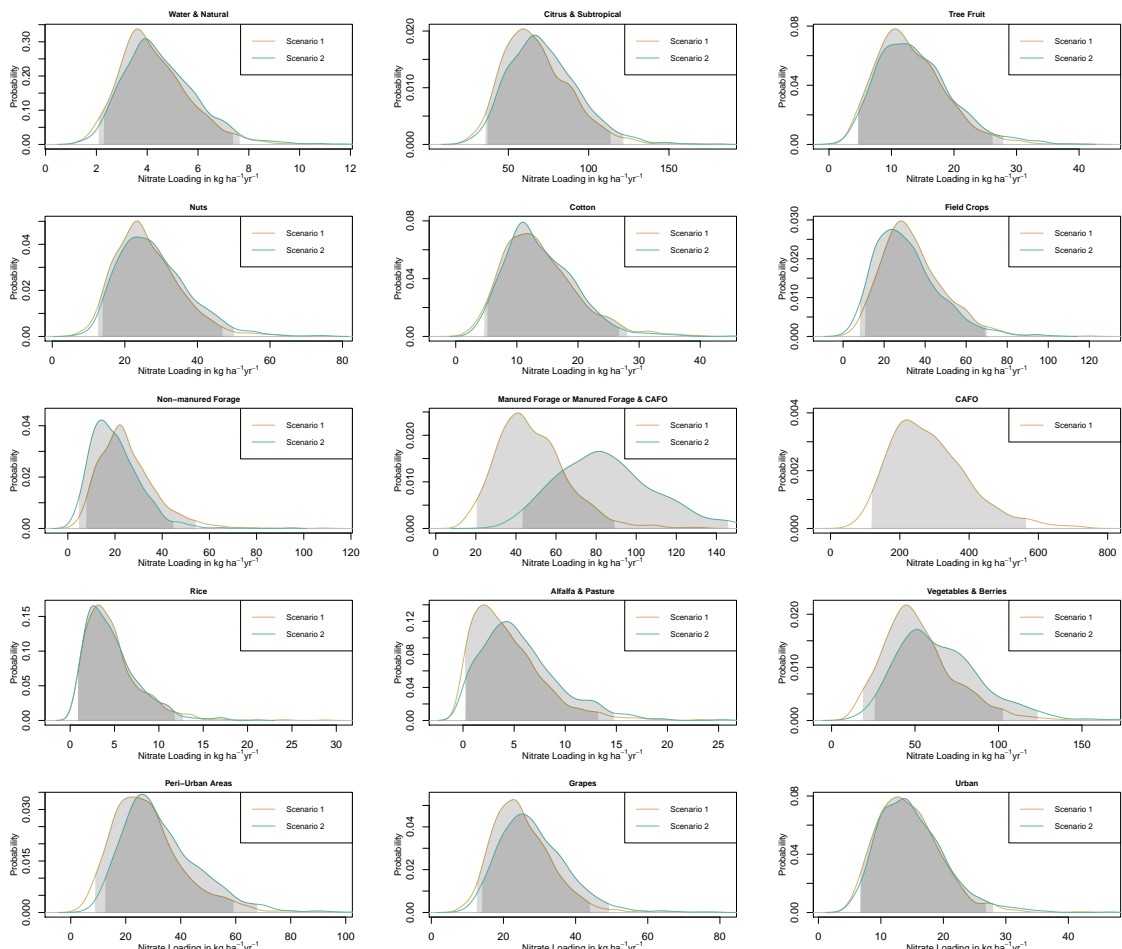

**Figure 5.** Posterior probability densities of estimated nitrogen loading for scenario 1 (tan) and 2 (teal). 95% Credibility intervals are represented by the light grey shading (dark grey shading occurs where scenario 1 and 2 estimates overlap).





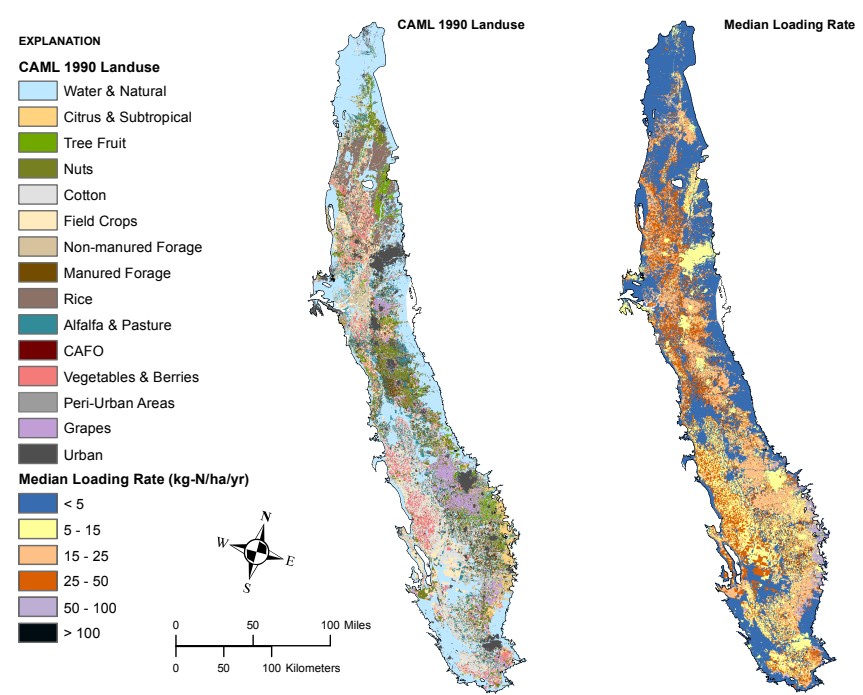

**Figure 6.** CAML landuse for the 15 landuse groups in scenario 1 (left side) and the same landuse groups keyed to the median estimated nitrogen loading rate in kg N ha$^{-1}$ yr$^{-1}$ for the corresponding group (right side).





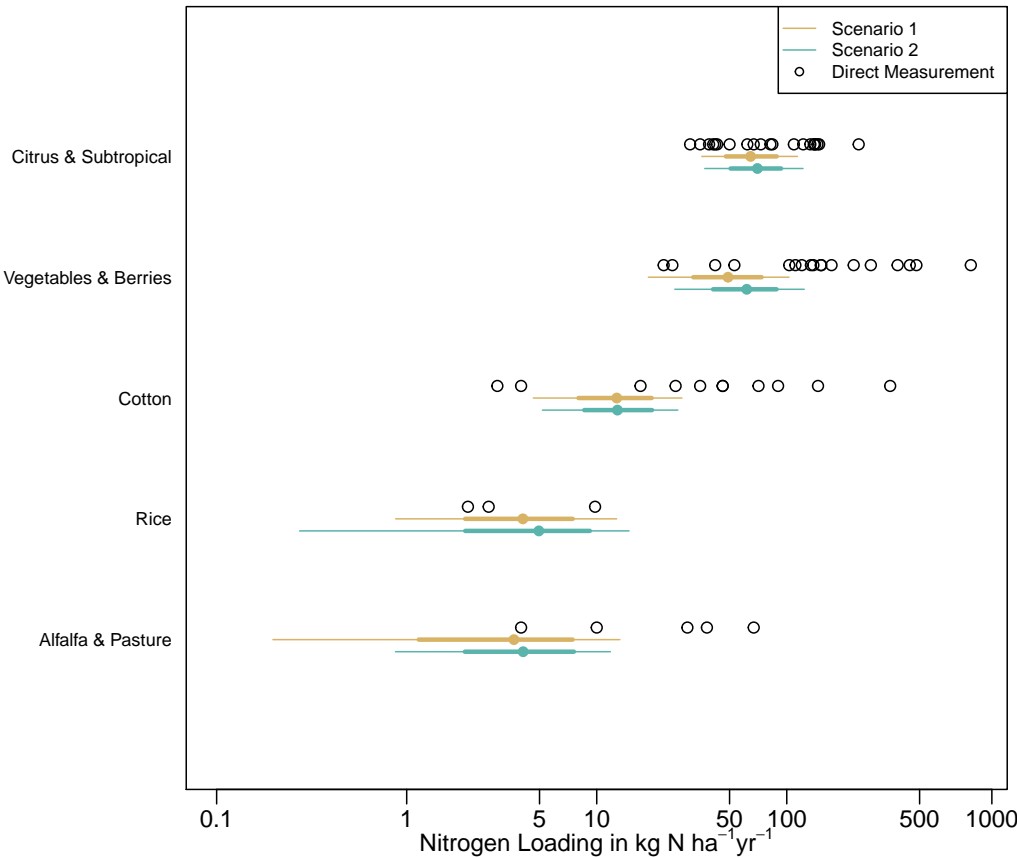

**Figure 7.** Credibility intervals of posterior probability densities of estimated nitrogen loading rates for selected crop groups plotted with historical direct measurements of nitrogen loading from California ((Devitt et al., 1976; Embleton et al., 1979; Letey et al., 1977; Pratt et al., 1972; Pratt and Adriano, 1973; Adriano et al., 1972; Allaire-Leung et al., 2001; Liang et al., 2014). Thinner lines are 95% credibility intervals, thicker lines are 68% credibility intervals, the solid dot is the median estimated value, and the open black circles are historical field measurements.



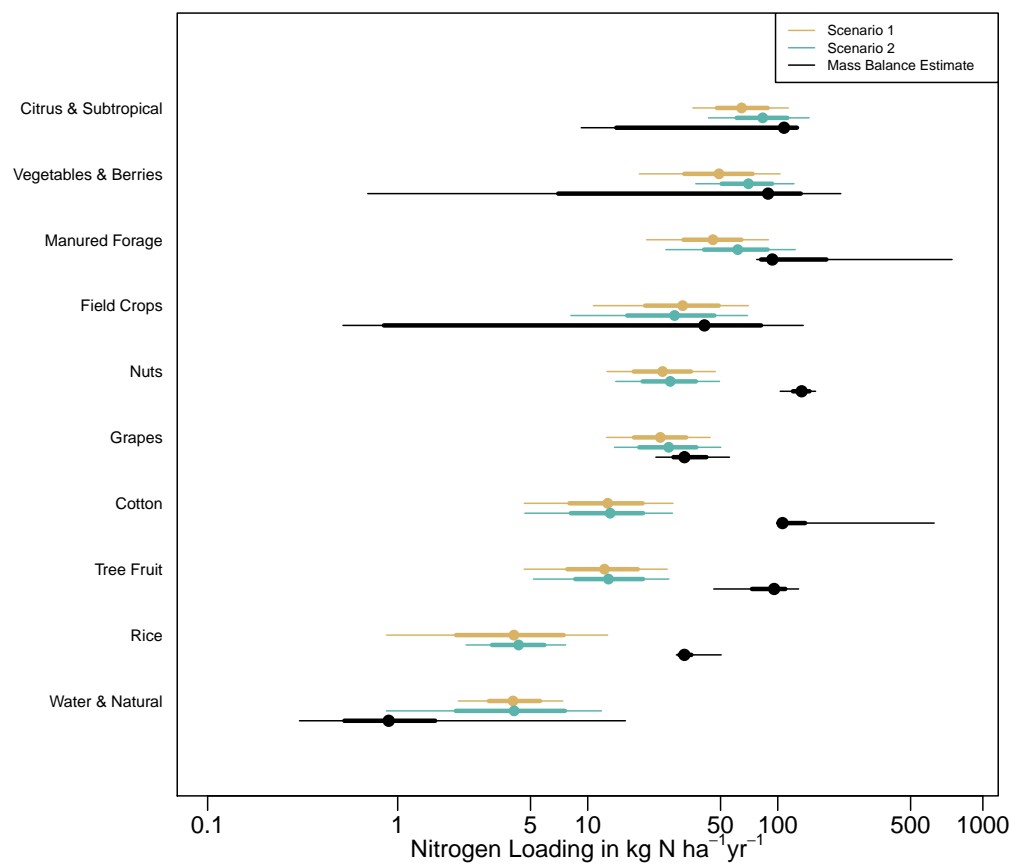

**Figure 8.** Credibility intervals of posterior probability densities of estimated nitrogen loading rates for selected crop groups plotted with the results from the Groundwater Nitrogen Loading Model (GNLM) (Viers et al., 2012; Rosenstock et al., 2013) (black). Thinner lines are 95% credibility intervals, thicker lines are 68% credibility intervals, and the dot is the median estimated value.



**Table 1.** Original data source, number of wells, and well location method for private wells included in final database (2149 wells total).

| Dataset Group | Dataset Subgroup | Number of Wells | Location Method/Accuracy |
|---|---|---|---|
| CASTINGS, CVRWQCB Fresno Office | Private wells on dairies | 361 | Located at reported coordinates of the dairy, the reported street address of the dairy, or the centroid of dairy parcel(s) (single, multiple adjacent parcels, or centroid of multiple non-adjacent parcels) (Boyle et al., 2012). |
| CASTINGS | GAMA Domestic Tulare County | 134 | Well locations randomly offset by 1/2 mile from true location (Boyle et al., 2012). |
| CASTINGS | Department of Pesticide Regulations (DPR) | 62 | Located at the centroid of the United States Public Land Survey System (PLSS) section (approximately 1 mi$^2$) in which the well resides (within 1/2 mile of the actual well location) (Boyle et al., 2012). |
| CASTINGS | Fresno County | 295 | Located at street address reported on well logs or centroid of the reported Assessor's Parcel Number (APN) (Boyle et al., 2012). |
| CASTINGS | The U.S. Geological Survey's (USGS) National Water Information System (NWIS) | 17 | Unknown (Boyle et al., 2012). |
| CASTINGS | Tulare County Environmental Health | 437 | Located at centroid of the reported APN (Boyle et al., 2012). |
| Lockhart et al. (2013) | None | 200 | Geographic coordinates digitized with imagery from Google Earth (Lockhart et al., 2013). |
| GAMA Domestic | Tehama, El Dorado, and Yuba Counties | 253 | Well locations randomly offset by 1/2 mile from true location. |
| CVRWQCB Rancho Cordova Office | Private wells on dairies | 390 | Geocoded using street address. |





**Table 2.** Median and 95% credibility interval bounds for estimated nitrogen loading rates by group for scenario 1 and 2.

| Scenario 1 Group | Median | Lower Bound | Upper Bound | Scenario 2 Group | Median | Lower Bound | Upper Bound |
|---|---|---|---|---|---|---|---|
| Water & Natural | 4.0 | 2.1 | 7.4 | Water & Natural | 4.3 | 2.2 | 7.5 |
| Citrus & Subtropical | 64.5 | 35.7 | 113.8 | Citrus & Subtropical | 69.0 | 37.0 | 116.5 |
| Tree Fruit | 12.2 | 4.6 | 26.2 | Tree Fruit | 13.0 | 4.7 | 26.2 |
| Nuts | 24.8 | 12.6 | 46.9 | Nuts | 26.7 | 13.8 | 46.8 |
| Cotton | 12.7 | 4.6 | 28.1 | Cotton | 12.7 | 5.3 | 25.8 |
| Field Crops | 31.6 | 10.7 | 70.1 | Field Crops | 28.8 | 9.4 | 71.4 |
| Non-manured Forage | 23.3 | 7.7 | 54.3 | Non-manured Forage | 18.1 | 4.4 | 44.9 |
| Manured Forage | 45.6 | 20.4 | 89.3 | Manured Forage & CAFO | 83.4 | 43.2 | 146.6 |
| Rice | 4.1 | 0.9 | 12.7 | Rice | 4.2 | 0.9 | 11.8 |
| Alfalfa & Pasture | 3.7 | 0.2 | 13.2 | Alfalfa & Pasture | 5.0 | 0.3 | 13.5 |
| CAFO | 268.9 | 118.5 | 565.1 | NA | NA | NA | NA |
| Vegetables & Berries | 49.1 | 18.7 | 102.8 | Vegetables & Berries | 61.4 | 25.7 | 122.5 |
| Peri-Urban Areas | 26.0 | 8.8 | 59.2 | Peri-Urban Areas | 30.1 | 12.3 | 68.6 |
| Grapes | 24.1 | 12.6 | 44.1 | Grapes | 27.1 | 14.0 | 49.3 |
| Urban | 13.9 | 6.6 | 26.8 | Urban | 14.2 | 6.5 | 27.9 |

NA: Not applicable.