# Peer review of "A Bayesian Approach to Infer Nitrogen Loading Rates from Crop and Landuse Types Surrounding Private Wells in the Central Valley, California"

_Hydrology and Earth System Sciences, 2017_

## Referee Comment (RC1) · Anonymous Referee #1 · 22 Jan 2018

The paper deals with a smart way of estimating nitrogen loading rates by an inference model. The topic is timely, quite original and suitable for HESS. This manuscript is a revised version of a previously submitted paper.

The manuscript requires some additional discussion on: 1. the model requires linearity of the transport processes. This is a limitation of this kind of approach that should be clearly stated. Therefore, the provided loading rates are loading rates at the ground-water surface, assuming that denitrification in the saturated zone can be neglected (probably true as discussed by the authors). All non-linear processes such as pro-
cesses linked to microbial activities in the vadose zone (denitrification, mineralization) are not taken into account. Therefore, this approach does not provide the loading rate depending on landuse and crops but depending on landuse, crops and kind of soil. The authors only addressed partly this point (see for example L5, p 11), neglecting the effects of the soil properties to nitrate propagation and transformation. 2. the definition travel time (or age) is unclear to me (eq. 1). It seems that it considers travel time in the aquifer only. What about the travel time in the unsaturated zone (which can be significant)? 3. Is it possible to distinguish effects of runoff compared to effect of storage in the vadose zone?

Minor: L 25, p. 7 Darcy's law instead of Darcy's Law. Avoid mixing units in the manuscript like feet (L25, p3) and meters (L26, p6 for example).

Therefore, I suggest moderate revision.

---

## Author Comment (AC1) · 24 Jan 2018

We appreciate the review from referee #1 and the constructive comments, all of which we would be able to address in a revised manuscript, as proposed here:

Point 1). The model indeed requires linearity in the transport process and this is a good point to clarify in the methodology. Uncertainties about nitrogen transformations in the vadose zone are implicitly embedded in the statistical approach, being one source of uncertainty. We propose a) to clarify that "recharge" is indeed the point in time when

nitrate mass enters the groundwater (not when it enters or leaves the root zone); b) to emphasize after equation (2) that the model implicitly assumes linearity in the transport processes between the points of recharge and pumping. c) add a specific point about uncertainty in nitrate fate and transport in soils and the deep vadose zone at the end of the paragraph on p5 line 14.

We also want to clarify that the results are not soil-specific in the sense that soil type surrounding a well with a nitrate sample plays an explicit role. The PDFs of loading rates implicitly reflect, among others, the variability in soils found in the Central Valley. The PDFs are specific only to the range and spatial patterns of soils found across the Central Valley. They are also specific to the crop distribution and land management practices found in the Central Valley.

Our approach in fact distinguishes two groups of environmental conditions that also account for the thickness of the vadose zone and for soil type: The choice of Ground-water Protection Zones as a spatial zonation to distinguished between two vulnerability levels is embedded in the model through the indicator I\_i in equation 4. We believe this point is well explained in the manuscript (last paragraph on page 6).

Point 2). The travel time in equation (1) is the travel time in groundwater. This will be addressed with the changes to improve definitions of recharge and loading suggested for Point 1).

Regarding both Points 1) and 2): The discussion section of the paper contains extensive treatment of the model limitations and interpretation of results arising from denitrification and travel time, in the vadose zone and in groundwater (see pages 15 and 16).

Point 3). We propose to add a sentence to the Project Area description that clarifies the negligible slope (

the effect of runoff cannot be estimated.

We agree to address both minor points in a revision.

We appreciate the reviewer's efforts.

---

## Referee Comment (RC2) · Anonymous Referee #2 · 3 Feb 2018

As stated by the authors, the manuscript is keyed to the assessment of the type of information that available groundwater quality data can provide towards the identification of nitrate loading rates from the various crop types spread across the Central Valley in California. The authors resort to a Bayesian analysis framework. The manuscript is a resubmission of a previously assessed work and I can see the relevance of the study to HESS.

The manuscript is well written and up-to-the-point, so I see no particular reason to delay its acceptance. There are only a few minor points which I think should be clarified

before acceptance and doing so would require only a set of minor revisions.

A point is related to the statement about the degree of innovation. The authors state that they develop "an innovative statistical framework". Then, they state that they "are not aware of any study that has employed Bayesian methods to estimate nitrate loading rates to groundwater.". They then conclude by saying that "The novel Bayesian tool developed here ...". I understand that it might be semantics, but I do think that the work is cast in the statistical framework of Bayesian analysis. As such, the authors do not develop an innovative statistical framework. Otherwise, they employ an existing framework and take advantage of it to solve a problem in an interesting way. Indeed, they then propose a new model within this general theoretical framework. This is my view, which seems to be in line with authors' concluding remarks, and can be as debatable as the view of the authors, of course. I simply think the terminology should be clarified.

The authors state that "Records in the database collected between the years 2000 to 2015 were selected. Locations with data collected in multiple years were assigned the median nitrate value of all the recorded measurements in order to prevent multiple samples of the same well and associated landuse.". I do concur with the approach. I am not sure if this course of action is somehow masking temporal dynamics of concentrations in the groundwater system. I am not sure there is an easy answer to this and I would just like to have the authors' idea on this aspect.

I am not sure if the authors are explicitly considering measurement data uncertainty in their Bayesian framework, but I might have missed the information. In any case, I think that some information about this could be provided.

I understand the reasoning behind the use of Student t-distribution (or log-normal distribution for potential recharge rates) priors and I do agree with it. Did the authors try to consider other formats of priors and obtained the same results?

I understand that the authors finally employ "a total of 1,000 realizations per parameter". Is the shape of the resulting target densities (or their key moments) depending

strongly on the number of realizations retained? It also seems that the number of realizations retained is relatively high with respect to the total number of realizations produced and some comments on this could be beneficial, to provide guidance to future users in other settings.

---

## Author Comment (AC2) · 20 Feb 2018

We appreciate the comments from referee #2. Below is a point by point discussion:

Point 1) We understand the point of view of the referee and agree we employed an existing statistical framework (Bayesian methods). Perhaps the use of the term "statistical framework" to describe our methods is not appropriate here and "innovative use of existing Bayesian methodology" and "the novel approach" or something similar would be more accurate. We would be willing to clarify the related statements in a revision.

[Figure]

Point 2) We think this is an excellent observation for the referee to point out and agree that the approach could be masking temporal dynamics of nitrate concentrations in groundwater. For example, of the initial set of wells in the dataset (before random location sampling) about 30% were sampled multiple times. Of those wells with multiple samples, the median range of observed nitrate values was 3.40 mg/L NO3-N (direction independent). Therefore, for this particular dataset, there are likely some significant temporal patters that could be investigated. In this paper, however, we focused on long-term average N loading from crop groups. For that, we maximized the number of samples available for the Bayesian analysis and did not split wells into temporal groups. The temporal aspects of nitrate loading are beyond the scope of this paper and may be best addressed as a separate study. The results of our study therefore represent the median rate of each landuse or crop groups between 2000-2015. We will add a sentence or two in the methods section to clarify this topic.

Point 3) We did not incorporate measurement uncertainty into our analysis here. Nitrate measurements for wells in the study database are from multiple agencies and laboratories. Uncertainty will vary between laboratories, analysis and field methods. However, these uncertainties are typically very small, especially compared to the concentrations at which nitrate becomes a concern (greater than 5-10 mg/L NO3-N). For example, field duplicates were collected for a subset of 200 wells in this current study (20 field duplicates) and the field duplicates had an average percent difference between samples of about 0.50 % (Lockhart et al. 2013). Laboratory uncertainty for internal laboratory duplicates for the same set of wells was similarly low, with many measurements having a difference of 0.00 mg/L NO3-N. We agree to add a brief explanation to this effect in the methods section.

Point 4) We experimented with several versions of this model including versions with a fixed recharge rate, various landuse/crop groupings, with/without the attenuation factor, and various likelihood and prior distributions. Results were relatively stable across model versions (assuming the choice of likelihood and prior were reasonable), with the

relative contributions of each landuse or crop group remaining approximately the same. While we did not make formal comparisons of various model results, we observed that the recharge rate parameter had the most dramatic effect on the model predictions and therefore, we decided to add it in to the model as a variable parameter.

Point 5) After the model "burn-in" period, each MCMC chain was sampled 200,000 times, with a thinning interval of 400 (every 400th sample was retained for final analysis). We feel this is a relatively low number of realizations to keep (per chain) compared to the number of samples in the chain (only 0.25 percent of MCMC samples were retained per chain, per parameter). This in in an effort to reduce the amount of autocorrelation between the MCMC samples. In order to determine the chain length and thinning interval were adequate, we plotted autocorrelation plots with the mcmcplots function in the R package mcmcplots (Curtis et al., 2015), for each chain for each parameter. Autocorrelation plots indicated low autocorrelation for each parameter at the indicated chain length and thinning rate. In addition, running mean plots indicate a convergence of the distribution means for the two MCMC chains (for each parameter) after approximately 500 samples. The choice of the chain length and thinning rate are highly dependent on the specific application, and we recommend others analyze the traceplots, autocorrelation plots, and running mean plots such as the ones produced by the R package mcmcplots.

References K.M. Lockhart, A.M. King, and T. Harter, 2013. Identifying sources of groundwater nitrate contamination in a large alluvial groundwater basin with highly diversified intensive agricultural production. Journal of Contaminant Hydrology 151 (2013) 140–154.

S. McKay Curtis, Ilya Goldin, Evangelos Evangelou, 2015. mcmcplots: Create Plots from MCMC Output, Version 0.4.2.

---

## Author Response (AR1)

**Changes in response to Reviewer #1 comments. Line and page numbers based on original unrevised submission.**

In response to point 1 regarding the uncertainty in nonlinear/microbial nitrogen processes:

Page 4 Line 21: Here we added the sentence: "Here, we assume linearity in the transport processes between the points of recharge and pumping as we do not model sorption or microbial processes such as denitrification or mineralization". To clarify that we are not modeling these processes explicitly.

Page 5 line 14 – 19: We added uncertainties about nitrogen transformations in the vadose and saturated zones as a fourth source of uncertainty.

In response to point 2 regarding the clarification of the travel time:

Page 4 Line 17: We added the phrase "as saturated zone travel time" to clarify that we are not considering vadose zone travel time here.

In response to point 3 regarding runoff to streams:

Page 3 Line 29: We have added the sentence: "Due to the low relief of the CV (slopes typically less than 0.2%), irrigation runoff to streams is generally considered negligible compared to recharge to groundwater".

Various changes:

Page 7 Line 25: We have removed the capitalization of the word "Law".

Project Area section: Converted ft to m, miles to km, and acres to hectares.

Page 8 Line 1: Converted ft to m.

Page 13 Line 13: Corrected acres to be hectares.

Table 1: Changed square miles to square km.

**Changes in response to Reviewer #2 comments. Line and page numbers based on original unrevised submission.**

Point 1 regarding wording of model descriptions and approach:

Page 2 Line 33: We have changed the phrase "innovative statistical framework" to "innovative use of existing Bayesian methodology".

Page 17 Line 5: We have changed the phrase "The novel Bayesian tool" to be "the novel approach".

Point 2 regarding temporal dynamics:

Page 6 Line 10: We added the phrase "In this study, we focused on long-term average nitrogen loading from each crop and landuse group".

Section 2 Paragraph 5: We have added the sentences "The groundwater data used in this study represent the average most recent (2000-2015) impact in domestic wells from the average nitrogen loading in each crop or landuse group at some time prior to sampling. Each domestic well water sample represents a mix of water age (Horn and Harter, Groundwater, 2009; Alikhani et al, J of Hydrology, 2016, Eberts et al, Hydrogeology J, 2012). ".

Point 3 regarding measurement uncertainty:

Page 6 Line 14: We have added the following sentences to this line and split the paragraph into two. "Measurement uncertainty was not explicitly considered as a source of uncertainty in the model. Measurement uncertainty will vary between laboratories, analysis, and field methods. However, these uncertainties are typically very small (less than 0.50% difference between duplicates), especially compared to the concentrations at which nitrate becomes a concern (greater than 5-10 mg/L $NO_3$-N)."

Point 4: no changes necessary.

Point 5 regarding the choice of the burn-in and thinning intervals:

Page 9 Line 15: We have added the sentence "Trace, running mean, and autocorrelation plots were visually inspected to confirm convergence, proper mixing of chains, and adequate burn-in period and thinning interval. Such plots were made with the R package mcmcplots (McKay and Goldin, 2012)."

**Note to editor about additional changes**
We have updated some reference values to which we compare our study results (Harter et al., 2017). Our initial submission referred to an outdated model output from Harter et al. 2017, which has since been updated. As a result, we have updated some values in the discussion and relevant comments. In addition, we updated Figure 8 to incorporate the new values, and Figure 7 to match Figure 8 in regards to the vertical lines we added to aid with interpretation.

[revised manuscript text omitted]

20 N. Measurement uncertainty was not explicitly considered as a source of uncertainty in the model. Measurement uncertainty will vary between laboratories, analysis, and field methods. However, these uncertainties are typically very small (less than 0.50% difference between duplicates), especially compared to the concentrations at which nitrate becomes a concern (greater than 5-10 mg/L $NO_3$-N).

Well geolocation methods varied depending on the source of the data. When geographic coordinates (latitude and longitude)

25 of the private wells in the dairy monitoring program were not available, the wells were located using the dairies street address and placed at the centroid of a dairy's land parcels. The methods for locating the wells varied for each of the other data sources including geographic coordinates, geocoded addresses, offsets by a random small distance, United States Public Land Survey System (PLSS) section, and Assessor's Parcel Number (APN) (Table 1). Due to the well location methods, many wells had overlapping locations. Where multiple wells were geolocated to a single location, a single well was chosen at random to

30 represent that location. Wells outside of the alluvial aquifer system boundary were excluded from the analysis. The final nitrate database had a total of 2149 wells.

Intrinsic aquifer properties were evaluated as an indicator for additional risk for or protection from nitrogen contamination. Here we choose a simple binary indicator: California Department of Pesticide Regulations (CDPR) Groundwater Protection

Areas (GWPAs) are 1.6 km by 1.6 km square sections that are vulnerable to the leaching of pesticides and are defined by the following criteria: previous detections of pesticides in that section, contains coarse soils and a depth to groundwater less than 21 m, or contains runoff-prone soils and depth to groundwater less than 21 m (California Department of Pesticide Regulation, 2017a). These zones are either vulnerable to contamination due to non-point source leaching of irrigation water ("leaching

5 GWPAs") or direct flow paths through hardpan soils (ditches, dry wells, poorly sealed wells, "runoff GWPAs") (California Department of Pesticide Regulation, 2017b). The properties that lead to vulnerabilities from pesticide contamination - shallow depth to groundwater, short residence time in the vadose zone, low reactivity of aquifer sediments - also increase the possibility of nitrate contamination. Wells located within a GWPA zone were  hypothesized to be subject to an increased risk of nitrate contamination. The non-parametric Kruskal-Wallis statistical test was performed on the nitrate

10 values for wells in each of the two groups ( GWPA versus non-GWPA wells). The Kruskal-Wallis test is a ranked one-way analysis of variance which tests whether two groups of values should be considered independent or from the same distribution.

The groundwater data used in this study represent the average most recent (2000-2015) impact in domestic wells from the average nitrogen loading in each crop or landuse group at some time prior to sampling. Each domestic well water

15 sample represents a mix of water age (Horn and Harter, 2009; Alikhani et al., 2016; Eberts et al., 2012) . Ransom et al. (2017) showed that landuse and nitrogen leaching patterns from the 1970s are most closely associated with recent groundwater nitrate measurements. 
[revised manuscript text omitted]

NA: Not applicable.